# Understanding Alveolar echinococcosis patients' psychosocial burden and coping strategies—A qualitative interview study

Christoph Nikendei[1]*, Anja Greinacher[1], Anna Cranz[1], Hans-Christoph Friederich[1], Marija Stojkovic[2], Anastasiya Berkunova[2]

1 Centre for Psychosocial Medicine, Department of General Internal Medicine and Psychosomatics, Heidelberg University Hospital, Heidelberg, Germany, 2 Section of Clinical Tropical Medicine, Department Infectious Diseases, Heidelberg University Hospital, Heidelberg, Germany

* christoph.nikendei@med.uni-heidelberg.de

**Data Availability Statement:** Data cannot be shared publicly because the interview transcripts contain personal data including information that

## Abstract

### Background

Alveolar echinococcosis (AE) is a serious parasitic zoonotic disease that resembles malignancy with clinically silent infiltrative growth predominantly involving the liver. AE patients show high levels of comorbid psychological burden and fear of disease progression. This study aimed to examine AE patients' perspective on their disease-related psychosocial burden using qualitative methods.

### Methods

We conducted N = 12 semi-structured interviews with AE patients focusing on their disease-related psychosocial burden, coping strategies, information seeking behavior, and subjective illness concepts. To this end, AE patients from a previous quantitative cross-sectional study were invited to participate. After verbatim transcription, interviews were analyzed thematically.

### Results

After analysis, data was grouped into five main themes: A) Perceived disease-related burden, B) Coping with disease-related burden, C) Disease-related impact on their social environment, D) Facing the future with the disease, and E) Disease-related information seeking behavior and subjective illness concepts. All participants perceived AE as a severe disease with inextricably linked biological, psychological, and social effects. Key positive influences reported included the provision of information and access to informal and formal support, including the ability to lead active personal and professional lives for as long as possible. Self-directed, web-based information seeking often led to increased feelings of hopelessness and anxiety.

### Conclusion

Our findings underscore the need to consider psychosocial morbidity in AE patient management. To reduce psychological burden, address disease-related apprehensions, and to

may allow personal identification of participants. Request for access to data can be addressed to ethikkommission-I@med.uni-heidelberg.de (ethics committee of the University of Heidelberg).

**Funding:** The author(s) received no specific funding for this work.

**Competing interests:** The authors have declared that no competing interests exist.

prevent stigmatization, health professionals need to provide AE patients with comprehensive disease-related information to improve patient and social awareness.

## Author summary

Alveolar echinococcosis (AE) is a serious parasitic zoonotic disease that resembles malignancy with clinically silent infiltrative growth predominantly involving the liver. Semi-structured interviews were conducted with N = 12 AE patients focusing on their disease-related psychosocial burden, coping strategies, information seeking behavior, and subjective illness concepts. All participants perceived AE as a severe disease with inextricably linked biological, psychological, and social effects. Key positive influences reported included the provision of information and access to informal and formal support, including the ability to lead active personal and professional lives for as long as possible. Self-directed, web-based information seeking often led to increased feelings of hopelessness and anxiety. Disease-related apprehensions, and to prevent stigmatization, health professionals need to provide AE patients with comprehensive disease-related information to improve patient and social awareness.

## Introduction

Alveolar echinococcosis (AE) is considered the parasitic zoonosis associated with the highest morbidity in the northern hemisphere's temperate climates. The Echinococcus multilocularis' life cycle mainly passes between foxes and rodents [1]. Nowadays, the poor prognosis in humans has improved following earlier diagnosis with better diagnostic tools and disease management techniques [2]. However, apart from surgical en bloc resection at an early stage of the disease when the infection is still asymptomatic, curative treatment is still unavailable. Later treatment aims at suppressing further growth with parasitostatic pharmacotherapy (benzimidazoles) [1–3].

Despite its similarity to cancer and the disease's chronic, disabling nature, AE patients' psychosocial burden has received little attention so far. To our best knowledge, only two studies have examined comorbid psychological burden and quality of life in AE patients to date. Schmidberger et al. [4] examined health-related quality of life (QoL) in 30 AE patients. While the physical QoL did not differ between both groups, they found reduced mental QoL in AE patients compared to healthy controls, taking conservative drug or surgical treatment regimens in account [5]. In our previous quantitative study [6], we examined 47 AE patients using a cross-sectional design and found that their depression, anxiety, and somatic symptom levels exceeded norm sample means, while their physical quality of life was poorer in comparison. Moreover, fear of disease progression exceeded levels found in cancer patient samples. Both studies point to a need for routine psychosocial screening and psychotherapeutic support offers as part of AE patients' care management.

Resilience factors are understood to be personality traits and characteristics which help individuals to cope with burdens and can, thus, protect them from developing (comorbid) psychological disorders. Several biological, individual, and social resilience factors have been examined: sense of coherence [7], attachment style [8], coping style [9], hardiness and successful meaning-making [10], as well as social support [11]. In line with these observations, AE patients in Nikendei et al.'s study [6] who reported a secure attachment style showed less

psychological burden than patients reporting an insecure attachment style. However, until now, all studies on comorbid psychological burdens and resilience factors in AE patients have relied on quantitative psychometric measures.

To bridge this research gap, our study aimed to examine a subset of AE patients from a previous quantitative cross-sectional study by Nikendei et al. [6] with focus on their perceived disease-related psychosocial burden, coping strategies, awareness, information-seeking behavior, and subjective illness concepts using qualitative methods.

## Methods

### Ethics statement

All participants gave their verbal and written informed consent and could withdraw their participation without disadvantage. All evaluated data were anonymized. The study was approved by the Heidelberg University Hospital Ethics Committee (Ethics Application No. S-232/2016). The study was conducted in accordance with the Declaration of Helsinki (most recent version: Fortaleza, Brazil, 2013).

### Study design and study population

For this qualitative study, we invited a subsample of AE patients participating in the previous cross-sectional study of Nikendei et al. [6] to take part in semi-structured interviews. All participants had been previously diagnosed with AE [12] and were consecutively recruited during a routine AE check-up at the clinic for echinococcosis at the Section of Clinical Tropical Medicine, Heidelberg University Hospital, Germany. 12 of 57 AE patients seen for echinococcosis between June 2016 and May 2017 participated in in the qualitative study and were invited to be interviewed. There were no exclusion criteria. Request for access to data can be addressed to ethikkommission-I@med.uni-heidelberg.de (ethics committee of the University of Heidelberg).

### Sociodemographic data

The following sociodemographic data were collected: Gender, age, nationality, marital status, time since diagnosis, AE status. Regarding AE status, patients were classified into the categories 'critical', 'uncritical', and 'cured'. The following AE patients are defined as 'critical': AE lesion(s) unresectable, AE progression not well controlled by benzimidazoles, significant adverse benzimidazole effects and/or complications, such as cholangitis, biliary system obstruction, compression of blood vessels or adjacent organs, or distant metastases. 'Uncritical' contains unresectable AE lesion(s), but AE progression well controlled by benzimidazoles. 'Cured' are patients free of disease at long term follow-up, after resection of AE lesion(s) and two years of postoperative benzimidazoles. In addition, data regarding ongoing psychotherapy and/or psychopharmacological treatment, as well as AE-related sick leave or early retirement were recorded.

### Semi-structured qualitative interviews

Following Helfferich's methodological approach [13], we designed a semi-structured interview guideline (see Table 1) that included key questions followed by probing questions; further clarifying questions were added, if necessary. Due to the specific research questions and the approach of AE patients' consecutive recruitment during routine AE check-ups, a methodological approach using half-standardized individual interviews was chosen. A Heidelberg University Hospital PhD student (AB) conducted the interviews via telephone or in person after a

**Table 1. Interview guide with the most important questions used in the semi-standardized interview.**

| Leading interview questions |
|---|
| How did you experience the diagnosis? |
| How did you experience the course of the disease? |
| How did you find out about the disease? |
| How do you think you contracted the disease? |
| What disease-related burdens do you experience? |
| • Physical |
| • Psychological |
| • Social |
| • Partnership |
| • Familial |
| How do you cope with these burdens? |
| Who have you told about the disease? |
| Do you feel stigmatized because of the disease? |

routine AE check-up at the Department of Echinococcosis of the Section of Clinical Tropical Medicine, Heidelberg University Hospital. The interviewer was specially trained in conducting semi-structured interviews and was supervised by an experienced senior researcher (CN).

## Analysis

The descriptive demographic variables and baseline characteristics were analyzed with the Statistical Package for the Social Sciences (SPSS) program version 24 [14]. The interviews were digitally recorded and transcribed verbatim. Following Mayring's principles of qualitative content analysis [15], we used the software MAXQDA [16] to help analyze the data. First, we identified single or multiple sentences as quotes, representing the most elemental unit of meaning [17]. Next, these statements were coded and summarized into relevant categories. We reviewed our coding system, categories, and coded statements by checking their coherence. After checking the coding system and codes for coherence, the categories were grouped into main themes. An independent researcher performed an inter-coder agreement check. Finally, we discussed the categories and main themes to reach consensus and adjusted them as necessary.

## Results

### Sample description

The interview participants' detailed sample characteristics are shown in Table 2.

### Qualitative interview results

We identified 130 individual codes and derived 17 categories, which were then grouped into five main themes: A) Perceived disease-related burden, B) Coping with disease-related burden, C) Disease-related impact on interaction with social environment, D) Facing the future with the disease, and E) Disease-related information seeking behavior and subjective illness concepts. Sample statements for each category within the main themes A) to E) are shown in Tables 3 to 5. The number of coded statements is indicated in parentheses.

**A) Perceived disease-related burden (52).** *A.1 Physical burden (7)*. The interviewees reported a wide range of disease-related, debilitating physical symptoms, including chronic fatigue, listlessness, reduced physical stamina, and decreased overall energy levels which impaired their everyday functioning and their quality of life.

**Table 2. Interviewed AE patients' sociodemographic characteristics (n = 12).**

| AE patients' characteristics | | |
|---|---|---|
| **Gender** | **n** | **%** |
| Male | 5 | 41.7 |
| Female | 7 | 58.3 |
| **Age** | **n** | **R** |
| Median (Range) | 50 | 61 |
| **Citizenship** | **n** | **%** |
| German | 11 | 91.7 |
| Croatian | 1 | 8.3 |
| **Marital status** | **n** | **%** |
| Married | 4 | 33.3 |
| In a relationship | 2 | 16.6 |
| Widowed | 3 | 25.0 |
| Single | 2 | 16.6 |
| Other | 1 | 8.3 |
| **Time since initial diagnosis [years]** | **n** | **R** |
| Median (Range) | 3 | 10 |
| **AE status** | **n** | **%** |
| Critical | 7 | 58.3 |
| Uncritical | 3 | 25.0 |
| Cured | 2 | 16.7 |
| **Psychotherapeutic treatment** | **n** | **%** |
| Current Psychotherapy | 1 | 8.3 |
| Psychotherapy in past | 2 | 16.6 |
| Psychiatric medication | 1 | 8.3 |
| No data | 1 | 8.3 |
| **AE-related sick leave / retirement** | **n** | **%** |
| No sick leave | 6 | 50.0 |
| Sick leave | 6 | 50.0 |
| Retirement | 0 | 0 |

AE clinical status: 'critical' = AE lesion(s) unresectable, AE progression not well controlled by benzimidazoles, significant adverse benzimidazole effects and/or complications, such as cholangitis, biliary system obstruction, compression of blood vessels or adjacent organs, distant metastases; 'uncritical' = AE lesion(s) unresectable, AE progression well controlled by benzimidazoles; 'cured' = patient free of disease at long term follow-up after resection of AE lesion(s) and two years of postoperative benzimidazoles

*A.2 Mental stress caused by the difficulty of finding a diagnosis (11).* The interviewees reported that a considerable amount of mental strain had been caused by the difficulty of finding a diagnosis for their symptoms. Many interviewees reported that a malignant tumor was often suggested as a differential diagnosis at the beginning of the diagnostic process. Consequently, the interviewed individuals were often faced with the existential fear. Discovering they actually had AE was initially a relief to them, as they felt they were no longer facing certain death.

*A.3 Mental stress due to lack of information about cause and treatment options (12).* Many respondents described how they often agonized over the question of whether they themselves were to blame for their illness or could even have prevented it by behaving more cautiously. Furthermore, particularly after initial diagnosis, the interviewees reported how little they had

**Table 3. Quotes for main themes A) Perceived disease-related burden and B) Coping with disease-related burden.**

---

***A) Perceived disease-related burden (52)***

A.1 Physical burden (7)
"I feel tired all the time—always a little tired, listless and I have no drive." *P05*
"The fact that my body just doesn't function the way I'd like it to anymore, that really gets me down. Just that it won't work the way I want it to anymore." *P01*

A.2 Mental stress caused by the difficulty of finding a diagnosis (11)
"Just that I, I thought I was going to die soon—Quite existential fears. You know, because even liver patients, so if that had been cancer and that would have been this liver cancer, I don't think I would have had long to live. " *P10*
"The worst thing was actually having to hear that you had liver cancer every day; that you didn't have long to live. That's when I literally opened the window on the eighth floor–I was surprised that the windows actually open up there—and if one were to jump and then it would all be over." *P02*

A.3 Mental stress due to lack of information about cause and treatment options (12)
"And that's what I kept asking myself, why me, why? With a hundred cases or so a year, I just don't know." *P03*
"And when I'm alone at home and I sometimes get very depressed and then think to myself: why did I get this, why me?" *P05*

A.4 Social burden (6)
"I think it's really affects him when I'm not doing so well. It also puts strain on our relationship." *P01*
"Yes, others have no idea what it is like. They take it so lightly, but it really weighs heavily on me." *P05*

A.5 Burden on professional development (3)
"I was sent to a health clinic for two weeks, to rehab and when I came back, I was just fired straight away." *P01*
"It definitely messed up my studies. That was a major blow in that respect." *P03*

A.6 Treatment related burdens (13)
"There's always this pressure to be compliant and take them dutifully in the morning and at night." *P06*
"That's been my biggest problem from the beginning and still is: I can't seem to ingest enough fat to reach the required level." *P12*
"If I didn't have to take my pills every morning and every night, I wouldn't even think about it." *P02*

***B) Coping with disease-related burden (20)***

B.1 Relief after receiving a clear diagnosis (7)
"When I heard that, that there was some medicine and if I could tolerate it okay, it was definitely better than cancer." *P02*
"Yeah and then with the fox-tapeworm, once that was determined that it was fox-tapeworm, well, first of all it was a relief; because with the cancer, well, that would have been it—the absolute worst case." *P10*

B.2 Using leisure activities and work as resources (7)
"Yes, until then, I tried to block it out. I don't know; I think when you get a diagnosis, like, that you kind of push it away a little bit for self-protection because it's actually hard at that age. " *P03*
"So actually I don't deal with it at all. I've completely blocked it out of my mind." *P02*

B.3 Working on their inner attitude towards the disease (4)
"Yes it was a hard blow; but then I thought: I must try and find a way to deal with this myself–me helping me through my inner attitude after I had calmed down after, let's say, a week." *P08*
"I don't think about it that often and when I see that other people maybe more sick. When I see my brother who has cancer and he is also so positive and lives and laughs a lot and maybe that he has this positive attitude. That's a way of deal with this." *P04*

B.4 Seeking support from their family and social environment (2)
"My partner supports me tremendously." *P01*
"Yeah, support from my family, definitely; I have two kids, and that's been good." *P09*

---

known about their own disease and the available treatment options. Some described that when they had initially sought information, they had been misinformed about the nature of their disease and some had believed it was terminal at first.

*A.4 Social burden (6).* The interviewees also reported immediate disease-related negative effects on their social relationships. Their family, friends, and partners were also highly affected by their illness because they vicariously witnessed its physical and psychological burden. The interviewees also reported that their sometimes depressed mood and listlessness often led to conflicts with their loved ones. Some interviewees felt insufficiently supported and expressed a desire for more help and understanding from their social environment.

**Table 4. Quotes for main themes C) Disease-related impact on their social environment and D) Facing the future with the disease.**

| |
| --- |
| ***C) Disease-related impact on their social environment (14)*** |
| C.1 Being open about the disease (8) |
| "I'm very open about my illness, and I also tell everyone what kind of illness I have and what my condition is and what the situation is. Everyone at work knows, and all my friends know, too." *P10* |
| "So, I'll say, I don't exactly put a poster on the wall of my house, but my family knows about it, of course. And most of the people I hang out with know." *P06* |
| C.2 Reactions from their environment (6) |
| "Many people have asked if it's contagious. Then I said, as far as I know, it's not contagious, something can only happen in my body, I can't infect you." *P04* |
| "It does feel like that one or two might not want to hug you as much anymore, but you can't really think about that too much." *P11* |
| ***D) Facing the future with the disease (24)*** |
| D.1 Optimistic view of the future with the disease (10) |
| "And then I thought, I have to be positive about this. To keep smiling and keep on going." *P08* |
| "And I try to be very relaxed about things, I don't drive myself crazy. That's my attitude. And if something bad comes my way, I'll know it when I get to it." *P07* |
| D.2 Experiencing fear of progression (14) |
| "But as long as I can tolerate the tablets, that's fine. I'm just afraid that with the tablets, it's a little like chemo, that they will damage other organs at some point now." *P02* |
| "I have decided to have an operation because this permanent business of taking medication, that is something that I don't like at all. And I'm also afraid that I won't be able to tolerate the medication at some point." *P11* |

*A.5 Burden on professional development (3)*. The interviewees experienced significant restrictions in their professional and educational development because of the disease, especially due to extended sick leaves, reduced working hours (and thereby their financial resources) or job loss.

*A.6 Treatment related burdens (13)*. The interviewees experienced AE treatment as a significant burden. Adhering to the medication routine for so long (twice daily at the same time and with a lot of fat) required a great deal of effort and self-discipline. Furthermore, the medication's effectiveness is directly dependent on taking it correctly. Medication related side effects, such as hair loss, contributed to their distress. A smaller proportion of respondents reported experiencing no disease-related burden.

**Table 5. Quotes for main theme E) Disease-related information seeking behavior and subjective illness concepts.**

| |
| --- |
| ***E) Disease-related information seeking behavior and subjective illness concepts (20)*** |
| E.1 Subjective illness concepts (6) |
| "And of course, I also think: Did I catch it from my dog—from my previous dog? Because they think I caught it about 15 or 10 years ago. The dog and I were very close and that—what are you supposed to do? Really, you should always wash your hands. Well, I didn't always do that." *P11* |
| "I used to be out and about all the time, collecting blueberries; I often walked through the meadows with my dog. According to doctors, it would have come from that. But the exact cause it is no longer traceable. It will have happened when I was young." *P02* |
| E.2 Information sources used to gather information about the disease (9) |
| "I researched the Internet. I even picked out doctoral dissertations and read them, researched them on the Internet." *P11* |
| "Probably, the usual thing: Internet expertise. I think it was the Robert Koch Institute; one of the institutes had some quite good information; then, just the pages of universities, like Wurzburg, Ulm." *P06* |
| E.3 Psychological impact of disease-related information (5) |
| "I logged on to a forum, looked on the Internet, and the first sentence that I saw in there on the Internet about fox tapeworm was: fox tapeworm patients die a painful death. I will never forget that sentence." *P12* |
| "Yeah, so I googled a little, yes, but then also I stopped because that drove me crazy." *P09* |

**B) Coping with disease-related burden (20).**   *B.1 Relief after receiving a clear diagnosis (7).* Most of the respondents reported that they had been given a malignant tumor disease diagnosis at the beginning of the medical diagnosis process. After spending time in fear of having cancer, they experienced the clear diagnosis as great relief, as AE is associated with a better prognosis than liver cancer.

*B.2 Using leisure activities and work as resources (7).* The respondents mentioned several leisure activities (e.g. meditating), distraction (e.g. professional work), self-soothing techniques (e.g. self-instructions), as important resources for coping with disease-related stress. Some interviewees reported that they would actively try to avoid thinking or talking about their illness as much as possible.

*B.3 Working on their inner attitude towards the disease (4).* Many interviewees emphasized that they tried to work on a confident and positive attitude towards life believing that this would help them deal with the disease and contribute to their mental well-being.

*B.4 Seeking support from their family and social environment (2).* The respondents cited support from their family, especially partners and children, and social environment as another essential resource for coping with psychological stress.

**C) Disease-related impact on interaction with the social environment (14).**   *C.1 Being open about the disease (8).* Most respondents indicated that they talked openly about their disease with their friends and family. However, the degree of openness varied considerably. While some interviewees were more reserved, many respondents reported that they had also informed their employer and work colleagues about their disease.

*C.2 Reactions from their environment (6).* Some respondents felt burdened by the lack of understanding of their disease which was as contradictory to their own experience. Others feared that their illness could lead to feelings of disgust or rejection by others. Sometimes they were confronted with the question of whether their disease was contagious. They would continue to experience uncertainty and avoidance from their immediate environment despite having tried to inform and reassure them. Furthermore, they described how some people remained openly skeptical, even after their explanations, as to whether the affected respondents might not pose a direct risk of infection to them.

**D) Facing the future with the disease (24).**   *D.1 Optimistic view of the future with the disease (10).* Despite the risk of progression or relapse of their disease, most respondents had an optimistic view and believed their health would remain stable and did not fundamentally expect their condition to deteriorate.

*D.2 Experiencing fear of progression (14).* However, many respondents also perceived the fear of disease progression or not reaching the therapeutic Albendazole goal resulting in lost treatment efficacy and progressed liver lesions respectively. Respondents were particularly concerned about their remaining lifespan Some of the respondents were worried about developing drug intolerance or suffering permanent damage from the medication's side effects leading to the surgical resection of the affected liver segment so that they could stop their medication.

**E) Disease-related information seeking behavior and subjective illness concepts (20).**
*E.1 Subjective illness concepts (6).* While some respondents described close contact with their pets, i.e. dogs or cats, as cause of their initial infection, others considered their careless outdoor behavior, such as picking berries and wild garlic, as a possible starting point for transmission.

*E.2 Information sources used to gather information about the disease (9).* The respondents reported using a variety of sources to learn about their disease. Most often, respondents reported resorting to the Internet via search engines or scientific publications. The respondents also reported reading patient information from their treating physicians or tropical medicine institutes.

*E.3 Psychological impact of disease-related information (5).* As stated above, most of interviewees tried to find information on AE via the Internet. Most interviewees felt that the retrieved information underlined the disconcerting nature of the disease emphasizing the serious, often fatal prognosis. The interviewees reported feeling frustration and increased anxiety after their search which led them to now generally avoid further information if they could.

## Discussion

Using semi-structured interviews, we aimed to gain insight into AE patients' personal experience of their disease. We focused on AE patients' disease-related burden, coping strategies, perceived stigma, information seeking behavior, and subjective illness concepts. Following qualitative content analysis, we thematically grouped the data into five main themes: A) Perceived disease-related burden, B) Coping with disease-related burden, C) Disease-related impact on their social environment, D) Facing the future with the disease, and E) Disease-related information seeking behavior and subjective illness concepts.

With regard to theme *A) Perceived disease-related burden*, interviewed AE patients reported considerable AE-related psychosocial burden naming fatigue, listlessness, and general loss of energy as chief complaints, which were associated with significant limitations and impairment in their everyday life. Most of these complaints can be classified as depressive symptoms [18]. Since depressive symptoms are prominent in the interviewees' narratives, the results suggest that depressive symptoms must be understood as the most distressing psychological aspects of AE disease. In line with findings in other physically affected patients, such as cancer patients [19], AE patients show more somatic depressive symptoms than cognitive-emotional depressive symptoms. From depressive symptoms, chronic fatigue syndrome must be distinguished that is very common in cancer patients [20] and could also be an important direction for further research efforts in AE patients. This directly points to psychosocial interventions that have to be provided and tailored to AE patient's needs, which are discussed below.

Patients suffering from somatic diseases and from comorbid mental illness as well, show lower somatic treatment adherence [21–24], poorer recovery outcomes [25], suffer from higher mortality rates [26–28], and produce increased health care costs [29]. Furthermore, burdensome somatic symptoms have been shown to increase 2- to 3-fold in patients with depressive and anxiety disorders [30,31]. Our findings underline that effective AE treatment must involve a multidimensional approach addressing both the psychosocial and physical components. The reported somatic depression symptoms suggest that psychological treatment in terms of psychotherapeutic and psychopharmacological interventions should be considered. Psychotherapeutic interventions have to address coping strategies, promotion of (pre-)existing key resources, and the enhancement of self-efficacy, while recognizing and acknowledging somatic restrains, reflecting on interpersonal conflicts arising from psychological and somatic symptoms, and facilitating family and social environment support. Short and efficient screening instruments that are easy to complete and provide immediate and reliable depression and anxiety scores could help identify patients in need of psychotherapy and lead to timely treatment recommendations [32]. Since the time between the initial symptoms and confirmation of the AE diagnosis is a particularly vulnerable interval, providing psychosocial and psychotherapeutic support seems to be especially important during this time. However, comorbid mental health symptoms are still under-recognized in AE patients. In our previous study [6], only one out of nine AE patients with depression had received adequate treatment with psychotherapy and/or antidepressants.

With AE patients looking for increased support from family members, their social system becomes overburdened over time. The importance of psychosocial support (e.g., in support

groups) not only for patients with severe somatic diseases themselves, but also for family members is well-known from cancer patients [33,34]. However, we are unaware of any interventions for AE patients and their relatives to date. Such interventions are urgently needed, as psychosocial family and social environment support are highly needed and desired by AE patients, and at the same time, family and social environment seems to be a source of stigmatization that is most feared. Our data clearly indicates that a family approach is necessary, including AE-related information for reducing stigmatization tendencies, facilitation of psychosocial support, and the exploration of the patient's family system for establishing helpful interpersonal relationships. Ensuring that the family system itself is supported may be a prerequisite for family members being able to provide AE patients with the support they need. Future research should explore the systemic aspects of AE disease and relevant interventions.

Data as well reveal, that the original AE transmission cannot be determined is another burdensome factor for AE patients, that raises the haunting question of whether patients themselves could have prevented the infection. Obtaining valid medical information about AE and its disease progression also contributed to disease-related fears and underscores the need for early intervention strategies. Similarly, the respondents saw AE medication adherence and AE medication side effects as major burdens in living with AE. Our data suggests that the treatment regimens might contribute to AE perceived diminished quality of life, that has been observed in a study by Schmidberger et al. [4] and our own previous research [6].

With regard to *B) Coping with disease-related stress*: AE patients reported receiving final confirmation of their AE diagnosis provided considerable relief, since they no longer had to worry about a cancer diagnosis. Besides this important fact, AE patients named leisure activities, stable daily routine, and professional or scholarly activities as key resources in dealing with their disease, e.g. by strengthening their self-efficacy. Accordingly, work-life limitations were seen as extremely stressful. This implies that work-related activities should be maintained for as long as possible. These aspects have to be an indispensable part of psychotherapeutic topics and interventions as, outlined above.

The interviewees reported taking a positive approach to their disease and speaking openly about it, as outlined in theme *C) Disease-related impact on their social environment*. This is consistent with the literature, which shows that psychosocial support is a highly relevant resilience factor [11]. Nevertheless, they experienced questions about the risk of a potential infection as most stressful and observed inhibitions of individuals to touch them. These interpersonal processes have also to be addressed and accompanied during the psychotherapeutic process. It has been shown that if stigmatizing behaviors are not consistently confronted in social environments, depression may be exacerbated [35]. Stigmatization in patients suffering from infectious diseases is widely observed, and can have an important impact on mental health [36], and contribute to a lower health-related quality of life [37]. Therefore, it is important to explore suitable interventions for the reduction of stigmatization if needed. In patients with chronic diseases illness disclosure and use of coping strategies can be protective factors [38].

Theme *D) Facing the future with the disease* illustrates that the interviewees were anxious about prescribed AE medications losing their effectiveness and were burdened by fear of disease progression. Patients suffering from a chronic condition are often faced with uncertainty about the course of their disease and are more likely to develop increased threat expectation [39]. This is in line with our previous findings [6] in which AE patients' fear of progression of the disease was even higher than in cancer patient groups suffering from malignant melanoma or prostate cancer.

Theme *E) Disease-Related Information-Seeking Behavior and Subjective Concepts of Disease* shows that most interviewees attempted to obtain medical information by searching the

internet rather than consulting their physicians. A possible explanation for this could be that searching the Internet allows individuals more anonymity and protects them from anticipated disease related social stigmatization and shame [40,41]. A reluctance to discuss the disease and actively avoiding relevant subjects may explain why so few patients asked their physicians to clear up their concerns during their annual checkup appointments. On the downside, patients are more likely to be misinformed, more anxious and subsequently require more time and attention during consultations [42–44]. In regard to the patient-physician relationship, the treating physician's empathy and sensitivity to patients' individual needs and circumstances have been shown to be important characteristics, especially when it comes to information transfer and communication [45]. These findings clearly illustrate that experts are fully aware of the prominent feelings of shame, AE patients experience. Experts need to provide valid information, communicate this information sensitively and, in the age of social media, trust-worthy websites need to be made known or established as points of contact.

## Limitations

This study has several limitations. First, as is customary in qualitative research, it relies on self-report interviews. Accordingly, we cannot rule out a social desirability bias. Such a bias might even be more prominent, as interviews were conducted in a clinical setting where AE patients seek help for medical monitoring and check-up. Also the feelings of shame experienced by AE patients might have led to the detention of relevant information. Second, the sample size was rather small. We conducted 12 interviews in total, 4 of which were by telephone. However, literature suggests, that about 12 interviews are sufficient to reach content saturation [46], which was the case in the presented study. In order to learn more about differing socio-economic aspects and age on research topics such as internet use, further research is needed, as the presented sample size is too small to allow such subgroup orientated conclusions.

## Conclusions

Following the previous study by Nikendei et al. [6], our study assessed the psychological burden of AE patients using semi-structured interviews to gain an in-depth insight into aspects of psychosocial burden, resources, stigma, and fear of AE progression. The results show that the time until final diagnosis is extremely stressful, that exhaustion and fatigue symptoms are key complaints, and that it is imperative to identify burdened patients to provide psychosocial support and therapy. Multiple resources are available to AE patients, including maintaining AE patients in their work environment whenever possible. Additionally, in order to reduce fears of the disease and prevent stigma, more awareness needs to be raised within their family system and social environment as a whole. Future research should focus on chronic fatigue symptoms and family systemic implications for the course of the disease.

## Author Contributions

**Conceptualization:** Christoph Nikendei, Anja Greinacher, Anna Cranz, Hans-Christoph Friederich, Marija Stojkovic, Anastasiya Berkunova.

**Data curation:** Christoph Nikendei, Anastasiya Berkunova.

**Formal analysis:** Christoph Nikendei, Anastasiya Berkunova.

**Methodology:** Christoph Nikendei, Anja Greinacher, Anna Cranz, Anastasiya Berkunova.

**Writing – original draft:** Christoph Nikendei, Anastasiya Berkunova.

**Writing – review & editing:** Christoph Nikendei, Anja Greinacher, Anna Cranz, Hans-Christoph Friederich, Marija Stojkovic, Anastasiya Berkunova.

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
