## [Decision Letter · Decision Letter 0]

3 Apr 2023

Dear Prof. Dr. Nikendei,

Thank you very much for submitting your manuscript "Understanding Alveolar Echinococcosis Patients' Psychosocial Burden and Coping Strategies - A Qualitative Interview Study" for consideration at PLOS Neglected Tropical Diseases. As with all papers reviewed by the journal, your manuscript was reviewed by members of the editorial board and by several independent reviewers. The reviewers appreciated the attention to an important topic. Based on the reviews, we are likely to accept this manuscript for publication, providing that you modify the manuscript according to the review recommendations. 

Sincerely,

Francesca Tamarozzi

Section Editor

Francesca Tamarozzi

Section Editor

Reviewer's Responses to Questions

**Key Review Criteria Required for Acceptance?**

**Methods**

-Are the objectives of the study clearly articulated with a clear testable hypothesis stated?

-Is the study design appropriate to address the stated objectives?

-Is the population clearly described and appropriate for the hypothesis being tested?

-Is the sample size sufficient to ensure adequate power to address the hypothesis being tested?

-Were correct statistical analysis used to support conclusions?

-Are there concerns about ethical or regulatory requirements being met?

Reviewer #1: The manuscript is well written. The objectives of the study and the population are clearly described. The design seems appropriate and the sample size sufficient from the viewpoint of a clinician involved in the care of AE patients, but this should also be reviewed by an expert in the field of psychological research. I see no concerns regarding ethical or regulatory requirements.

Reviewer #2: The objectives of the study are clearly articulated. The study design is fairly straightforward. More rational should be provided about the chosen method (semi-structured interviews)- why was this appropriate for conduct a study of this kind? Further information should also be provided in the limitations of this method for understanding psychological wellbeing and how conducting interviews in a hospital setting may have shaped responses- this is not appropriately addressed currently. There are also some leading questions 'do you feel stigmatised' in the topic guides (table 1). More discussion of this should be provided and again this should be identified as a limitation.

**Results**

-Does the analysis presented match the analysis plan?

-Are the results clearly and completely presented?

-Are the figures (Tables, Images) of sufficient quality for clarity?

Reviewer #1: The analysis is mostly conclusive and clearly and completely presented. The tables are of sufficient quality and clarity.

Reviewer #2: The analysis is clear, however, it is a little deductive, and i would have expected to see more interconnection between the themes presented. For example, there is a clear link between the information provided in A2 and B1 that needs to be discussed together so that it is easy to the reader to see the bigger picture in the diagnosis communication pathway and how this shapes psychological wellbeing. C1/C2 could also be strengthened by considering these as forms of stigma as opposed to the themes currently presented- engaging with the stigma literature would support this part of the analysis. Finally, there is limited consideration of how socio-demographics shape the responses- adding this throughout would strengthen the results.

**Conclusions**

-Are the conclusions supported by the data presented?

-Are the limitations of analysis clearly described?

-Do the authors discuss how these data can be helpful to advance our understanding of the topic under study?

-Is public health relevance addressed?

Reviewer #1: The conclusions are supported by the presented data and relevant limitations to the study discussed. There is a great public health relevance as the topic of psychosocial burden and coping strategies in the field of AE is understudied. The presented analysis well reflects clinical experience in the care of AE patients.

Reviewer #2: The conclusions are supported by the data presented and this is an important study. There are a few areas of the discussion that should be revisited, including the section on depression. Currently, this is written from a very medicalised perspective (e.g. provide antidepressants), with limited exploration of how the personal and environmental factors presented in the results could also be useful to consider in intervention development. Additionally, family intervention is described as important, yet these are also the main points of stigma identified in the results. This should be considered/elaborated within the discussion.

**Editorial and Data Presentation Modifications?**

Reviewer #1: Due to the small sample size, I suggest using Median and Range instead of Mean and Standard Deviation in Table 2. Furthermore, the "AE Status" classification in Table 2 into "critical", "uncritical" and "cured" is uncommon and not defined in the methods section. I suggest classifying patients into the categories differently, i.e. "surgically cured", "inoperable - no biliary or vascular complications" and "inoperable - with biliary or vascular complications".

Reviewer #2: (No Response)

**Summary and General Comments**

Reviewer #1: As mentioned above the study is of high academic and clinical relevance and the manuscript is well-written.

Reviewer #2: This paper is important and will make a good contribution to the literature once the issues outlined above are addressed.

PLOS authors have the option to publish the peer review history of their article (what does this mean?). If published, this will include your full peer review and any attached files.

Reviewer #1: No

Reviewer #2: No

Figure Files:

Data Requirements:

Reproducibility:

References

---

## [Decision Letter · Decision Letter 1]

16 Jun 2023

Dear Prof. Dr. Nikendei,

We are pleased to inform you that your manuscript 'Understanding Alveolar Echinococcosis Patients' Psychosocial Burden and Coping Strategies - A Qualitative Interview Study' has been provisionally accepted for publication in PLOS Neglected Tropical Diseases.

Best regards,

Francesca Tamarozzi

Section Editor

Francesca Tamarozzi

Section Editor

Reviewer's Responses to Questions

**Key Review Criteria Required for Acceptance?**

**Methods**

-Are the objectives of the study clearly articulated with a clear testable hypothesis stated?

-Is the study design appropriate to address the stated objectives?

-Is the population clearly described and appropriate for the hypothesis being tested?

-Is the sample size sufficient to ensure adequate power to address the hypothesis being tested?

-Were correct statistical analysis used to support conclusions?

-Are there concerns about ethical or regulatory requirements being met?

Reviewer #1: (No Response)

Reviewer #2: Thank you for responding to the comments made. I believe that the points raised are now adequately addressed and the manuscript should proceed for publication. I would encourage the authors to continue work in this area whilst also engaging more with the qualitative methodological literature to understand the value of more in-depth approches to exploring mental wellbeing. Semi-structured interviews are likely to limit the depth of nuanced understandings. Further, I would encourage in the future more consideration of socio-demographic variables within the analysis- without these considerations it makes the use of these criteria in sampling seem somewhat meaningless. That said, i think the manuscript is greatly improved and worthy of publication.

**Results**

-Does the analysis presented match the analysis plan?

-Are the results clearly and completely presented?

-Are the figures (Tables, Images) of sufficient quality for clarity?

Reviewer #1: (No Response)

Reviewer #2: (No Response)

**Conclusions**

-Are the conclusions supported by the data presented?

-Are the limitations of analysis clearly described?

-Do the authors discuss how these data can be helpful to advance our understanding of the topic under study?

-Is public health relevance addressed?

Reviewer #1: (No Response)

Reviewer #2: (No Response)

**Editorial and Data Presentation Modifications?**

Reviewer #1: (No Response)

Reviewer #2: (No Response)

**Summary and General Comments**

Reviewer #1: My main review points have been adressed adequately.

Reviewer #2: (No Response)

PLOS authors have the option to publish the peer review history of their article (what does this mean?). If published, this will include your full peer review and any attached files.

Reviewer #1: No

Reviewer #2: No

---

## [Editor Report · Acceptance letter]

21 Jul 2023

Dear Prof. Dr. Nikendei,

We are delighted to inform you that your manuscript, "Understanding Alveolar Echinococcosis Patients' Psychosocial Burden and Coping Strategies - A Qualitative Interview Study," has been formally accepted for publication in PLOS Neglected Tropical Diseases.

Best regards,

Shaden Kamhawi

co-Editor-in-Chief

Paul Brindley

co-Editor-in-Chief
